# Homeostatic Model Assessment of Insulin Resistance for Predicting the Recurrence of Hepatocellular Carcinoma after Curative Treatment

**DOI:** 10.3390/ijms20030605

**Published:** 2019-01-30

**Authors:** Kenji Imai, Koji Takai, Tatsunori Hanai, Atsushi Suetsugu, Makoto Shiraki, Masahito Shimizu

**Affiliations:** Department of Gastroenterology/Internal Medicine, Gifu University Graduate School of Medicine, 1-1 Yanagido, Gifu 501-1194, Japan; koz@gifu-u.ac.jp (K.T.); hanai0606@yahoo.co.jp (T.H.); asue@gifu-u.ac.jp (A.S.); mshiraki-gif@umin.ac.jp (Mak.S.); shimim-gif@umin.ac.jp (Mas.S.)

**Keywords:** Hepatocellular carcinoma, recurrence risk, diabetes mellitus, insulin resistance, homeostasis model assessment-insulin resistance (HOMA-IR)

## Abstract

Diabetes mellitus (DM) is a risk factor for hepatocellular carcinoma (HCC). The purpose of this study was to investigate the impact of the disorder of glucose metabolism on the recurrence of HCC after curative treatment. Two hundred and eleven patients with HCC who received curative treatment in our hospital from 2006 to 2017 were enrolled in this study. Recurrence-free survival was estimated using the Kaplan–Meier method, and the differences between the groups partitioned by the presence or absence of DM and the values of hemoglobin A1c (HbA1c), fasting plasma glucose (FPG), fasting immunoreactive insulin (FIRI), and homeostasis model assessment-insulin resistance (HOMA-IR) were evaluated using the log-rank test. There were no significant differences in the recurrence-free survival rate between the patients with and without DM (*p* = 0.144), higher and lower levels of HbA1c (≥6.5 and <6.5%, respectively; *p* = 0.509), FPG (≥126 and <126 mg/dL, respectively; *p* = 0.143), and FIRI (≥10 and <10 μU/mL, respectively; *p* = 0.248). However, the higher HOMA-IR group (≥2.3) had HCC recurrence significantly earlier than the lower HOMA-IR group (<2.3, *p* = 0.013). Moreover, there was a significant difference between the higher and lower HOMA-IR groups without DM (*p* = 0.009), and there was no significant difference between those groups with DM (*p* = 0.759). A higher HOMA-IR level, particularly in non-diabetic patients, was a significant predictor for HCC recurrence after curative treatment.

## 1. Introduction

Hepatocellular carcinoma (HCC) is one of the most common cancers worldwide because more than half a million people are diagnosed with this malignancy yearly [1]. HCC generally develops in patients with liver cirrhosis due to persistent hepatitis B virus (HBV) or hepatitis C virus (HCV) infection, alcohol consumption, and immune-related hepatitis [2]. Active surveillance and watchful observation are used to diagnose HCC at an early stage in these high-risk patients. However, the prognosis of patients with this malignancy is very poor because only 46% of patients with HCC are diagnosed at an early stage, and most do not receive curative therapy [3,4]. Furthermore, even patients with HCC in whom curative therapy can be performed are extremely prone to recurrence; in fact, the 5-year recurrence rate after curative treatment has been reported to be more than 70% [5,6]. Therefore, in order to reduce the mortality of HCC, it is very important to identify significant predictors of HCC development and recurrence, as these factors are useful in the surveillance strategy.

In addition to the established risk factors such as HBV, HCV, and alcohol consumption, several components of the metabolic syndrome have been reported to be involved in the increased risk of HCC [7]. We have previously demonstrated that increases in the serum levels of leptin and oxidative stress and increased visceral fat accumulation, all of which are involved in obesity, predict the recurrence of HCC after curative treatment [8,9,10]. Among lifestyle-related diseases, diabetes mellitus (DM) is most deeply involved in liver carcinogenesis [7,11,12,13]. Obesity-related and diabetes-related metabolic disorders, such as impaired insulin sensitivity and non-alcoholic fatty liver disease (NAFLD)/non-alcoholic steatohepatitis (NASH), contribute to increasing the incidence of HCC with a non-viral etiology [14]. Importantly, a higher level of homeostasis model assessment-insulin resistance (HOMA-IR), an indicator of insulin resistance, is associated with an increased risk for recurrence of stage I HCC after curative treatment in HCV-positive patients [15]. These findings suggest that several indicators that are associated with metabolic syndrome, in particular, glucose metabolism abnormality, may be useful in assessing the recurrence risk for HCC after curative treatment.

In this study, we examined the impact of the complication of DM and the factors associated with glucose metabolism, including hemoglobin A1c (HbA1c), fasting plasma glucose (FPG), fasting immunoreactive insulin (FIRI), and HOMA-IR, on the recurrence of HCC in patients who received curative treatment for this malignancy. The purpose of this study was to determine whether these indicators can predict HCC recurrence after initial curative treatment.

## 2. Results

### 2.1. Baseline Characteristics and Laboratory Data

The baseline characteristics and laboratory data of the 211 patients (148 men and 63 women, average age 70.6 years) and the comparison of the groups with (*n* = 71) and without DM (*n* = 140) are shown in Table 1. These data were collected immediately before initial treatment. All the diabetic patients in this study were classified as having type 2 DM. HBV-negative and HCV-negative HCC was observed more often in patients with DM than in those without DM (*p* < 0.001), and patients with DM had a significantly higher body mass index (*p* < 0.001), HbA1c level (*p* < 0.001), FPG level (*p* < 0.001), HOMA-IR level (*p* = 0.047), and serum triglyceride level (*p* = 0.003). Serum total bilirubin levels were significantly lower (*p* = 0.022) but serum albumin levels were significantly higher (*p* = 0.028) in patients with DM than in those without DM. In addition, the frequencies of hypertension (*p* < 0.001), hyperlipidemia (*p* = 0.007), and NAFLD/NASH (*p* < 0.001) were significantly higher in patients with DM than in those without DM. There were no significant differences in age, serum levels of alanine aminotransferase, the platelet count, Child–Pugh score, first treatment regimen, and tumor-related factors, such as tumor makers (AFP and PIVKA-II) and clinical cancer stage, between the groups with and without DM.

### 2.2. Impact of DM (Diabetes Mellitus) and Glucose Metabolism-Related Factors on Recurrence-Free Survival in Patients with HCC (Hepatocellular Carcinoma) after Curative Treatmen

The 1-year, 3-year, and 5-year recurrence-free survival rates of all enrolled patients were 63.0, 32.9, and 20.5%, respectively. As shown in Figure 1, there was no significant difference in the recurrence-free survival rate between the patients with and without DM (*p* = 0.144). There were also no significant differences between higher and lower levels of HbA1c (≥6.5 and <6.5%, respectively; *p* = 0.509; Figure 2a), FPG (≥126 and <126 mg/dL, respectively; *p* = 0.143; Figure 2b), and FIRI (≥10 and <10 μU/mL, respectively; *p* = 0.248; Figure 2c). However, the higher HOMA-IR group (≥2.3) had HCC recurrence significantly earlier than the lower HOMA-IR group (<2.3, *p* = 0.013; Figure 2d). Moreover, there was a significant difference between the higher and lower HOMA-IR groups when the patients did not have DM (*p* = 0.009; Figure 3a), as there was no significant difference when the patients had DM (*p* = 0.759; Figure 3b). Similar results were also obtained from the subanalysis of the 195 patients without NASH/NAFLD, which is highly prevalent in type 2 DM [11]. Specifically, the factors associated with a glucose metabolism disorders, such as DM (*p* = 0.092; Appendix A), higher HbA1c level (*p* = 0.179; Appendix A), FPG level (*p* = 0.115; Appendix A), and FIRI level (*p* = 0.284; Appendix A), were not significant predictors of HCC recurrence; only higher HOMA-IR level predicted recurrence (*p* = 0.015; Appendix A). Moreover, even in non-NASH/NAFLD patients without DM, higher HOMA-IR level had a significant impact on HCC recurrence (*p* = 0.011; Appendix A), whereas there was no significant difference in patients with DM (*p* = 0.686; Appendix A).

Table 2 summarizes the predictors of HCC recurrence. Univariate analysis revealed that higher body mass index (*p* = 0.049) and HOMA-IR level (*p* = 0.014) were significant risk factors of HCC recurrence. Multivariate analysis identified that higher HOMA-IR level is the only independent predictor of HCC recurrence in this study (hazard ratio, 1.485; 95% confidence interval, 1.030–2.139; *p* = 0.034). This finding was consistent with that of a previous report demonstrating that higher HOMA-IR level is an independent predictor of stage-I HCC recurrence after curative radiofrequency ablation treatment in HCV-positive patients [15].

## 3. Discussion

It is widely accepted that there is a strong relationship between DM and the development of HCC [7,11,12,13,16]. A meta-analysis revealed that DM is associated with an approximately 2.5-fold increased risk of HCC [7]. The presence of DM remained an independent risk factor of HCC after adjusting for alcohol consumption or viral hepatitis [7,13,17]. Several studies have also described synergistic interactions between DM and other HCC risk factors, such as viral hepatitis infection and alcohol consumption [12,18,19]. In addition, DM is closely linked to NASH that is involved in the development of HCC [11]. Endogenous and/or exogenous hyperinsulinemia, hyperglycemia, and chronic inflammation observed in diabetic patients may be involved in liver carcinogenesis [12]. In particular, insulin resistance and hyperinsulinemia play a critical role in the development of HCC because insulin can exert a potentially mitogenic effect by activating its receptor in both precancerous and cancer cells [20].

The results of the present study clearly demonstrated that a higher HOMA-IR level (≥2.3), which is commonly used for evaluating insulin resistance [21,22], can be used to predict the early recurrence of HCC, whereas the complication of DM, a higher HbA1c level (≥6.5), higher FPG level (≥126), and higher FIRI level (≥10) were not associated with recurrence. Moreover, non-diabetic patients with a higher HOMA-IR level had a significantly increased risk of HCC recurrence after curative treatment, but diabetic patients with a higher HOMA-IR did not. These findings suggest that insulin resistance may be critically involved in the recurrence of HCC, and that a higher HOMA-IR level is an effective and useful predictor of recurrence. However, we should pay careful attention to the accuracy of HOMA-IR, as it might be decreased in advanced diabetic patients because hyperglycemia induces the inadequate secretion of insulin and the homeostasis between fasting glucose and insulin levels is disrupted in these patients [21]. Therefore, HOMA-IR should be used as a predictor of HCC recurrence, especially when the patients have a slight glucose tolerance disorder or are in a pre-diabetic phase.

In this study, the complication of DM did not promote the recurrence of HCC. These findings may indicate that diabetic patients are not necessarily apt to have an increased risk of the development and recurrence of HCC when they undergo adequate DM treatment. Improvement of both hyperinsulinemia and hyperglycemia and other metabolic abnormalities associated with obesity might be an effective strategy for preventing the development of HCC in patients with metabolic syndrome [23,24,25]. Exercise and nutritional therapies are essential for improving hyperinsulinemia, insulin resistance, hyperglycemia, and obesity in patients with DM [26,27]. Moreover, some studies have suggested that antidiabetic drugs affect the development of HCC [12]. For instance, metformin lowers the incidence of cancers, including HCC, in patients with DM [28,29]. Thiazolidinediones, which lower insulin resistance without directly affecting insulin secretion, also significantly decreased the risk of liver cancer [30]. On the other hand, insulin secretagogues and insulin analogs may increase the risk of HCC in diabetic patients [31,32,33,34]. These reports, together with the present study’s results, may suggest that improvement of a lifestyle habit, such as exercise and nutrition, and usage of antidiabetic drugs, which lower insulin resistance, are effective ways to prevent the development and recurrence of HCC in diabetic patients.

This study has several limitations. First, it was a retrospective, single-center study and the sample size of HCC patients with DM was comparatively small. Second, it remains unclear whether adequate DM treatment, which improves both hyperglycemia and hyperinsulinemia, could reduce the risk of HCC recurrence in patients with DM because glucose metabolism was evaluated only once, immediately before initial treatment. These limitations might cause a bias or influence the results of this study. Therefore, long-term, prospective studies in a larger number of patients should be performed to confirm the findings and the possibilities obtained from the present study. Moreover, the possibility that hyperglycemia, in addition to insulin resistance, which was revealed in the present study, is involved in HCC recurrence cannot be denied. Hyperglycemia among diabetic patients can increase oxidative stress in the cells, which can induce cancer development [22,23]. Improvement of hyperinsulinemia and hyperglycemia as well as obesity-related metabolic abnormalities might be beneficial in suppressing the development and/or the recurrence of HCC regardless of DM.

It should be emphasized again that a rapid and abundant increase in the incidence of HCC with a non-viral etiology is a serious health care problem worldwide [35]. Obesity-related and diabetes-related metabolic disorders have a great impact on such liver carcinogenesis and, therefore, can be beneficial markers for identifying the high-risk patients with HCC.

## 4. Materials and Methods

### 4.1. Patients, Treatment, and Determination of Recurrence

The diagnosis of HCC was made from a typical dynamic-study finding of enhanced staining in the early phase and attenuation in the delayed phase using imaging modalities including enhanced ultrasonography, dynamic computed tomography (CT), and dynamic magnetic resonance imaging (MRI). The selection criteria for the initial treatments were determined according to the guidelines for HCC by the Liver Cancer Study Group of Japan [36]. Patients were thereafter followed up with on an outpatient basis to assess the levels of serum tumor markers such as alpha-fetoprotein (AFP) and proteins induced by vitamin K absence or antagonist-II (PIVKA-II), and imaging modalities such as dynamic CT and MRI were performed every 3 months. Recurrent HCC was defined as the imaging characteristics of HCC described above of distant lesions in order to exclude local recurrence. The recurrence-free survival time was defined as the interval from the date of the initial treatment to the date of the recurrence or December 2017 for recurrence-free survivors.

We diagnosed 415 patients as having primary HCC in our hospital from 2006 to 2017. Among them, we determined that 211 patients were curative and enrolled them in this study, and they all met the following criteria: surgical resection or radiofrequency ablation conducted for the initial HCC treatment; and the imaging modalities after initial treatment exhibited complete disappearance of the imaging characteristics of HCC described above.

All study participants provided verbal informed consent, which was considered sufficient, as this study followed an observational research design that did not require new human specimens. This study design, including this consent procedure, was approved by the ethics committee of the Gifu University School of Medicine (approval number 29-26).

### 4.2. Examination of Glucose Metabolism

In order to evaluate glucose metabolism, the levels of HbA1c, FPG, FIRI, and HOMA-IR were measured before curative treatment of HCC. We determined patients as having DM if they already underwent DM treatment or met the following criteria: HbA1c level ≥6.5%, FPG level ≥126 mg/dL, or random plasma glucose level ≥200 mg/dL [37]. HOMA-IR was calculated as follows: HOMA-IR = FIRI × FPG/405 [22]. We set the cut-off values for HbA1c, FPG, and FIRI at 6.5%, 126 mg/dL, and 10 μU/mL, respectively, based on the generally established standard values [37]. The cut-off value for HOMA-IR was set at 2.3, which predicted the recurrence of stage I HCC after curative treatment in a previous study [15].

### 4.3. Statistical Analysis

Baseline characteristics were compared using Student’s *t*-test for continuous variables or the χ^2^ test for categorical variables. Holm-Bonferroni correction was used to counteract the problem of multiple comparisons [38]. Univariate and multivariate analyses were performed using the Cox proportional hazards model to identify the risk factors of HCC recurrence. Recurrence-free survival was estimated using the Kaplan–Meier method, and differences between curves were evaluated using the log-rank test. Statistical significance was defined as *P* < 0.05. All statistical analyses were performed using R version 3.3.3 (The R Project for Statistical Computing, Vienna, Austria; http://www.R-project.org/).

## 5. Conclusions

A higher HOMA-IR level, an insulin resistance marker, is associated with HCC recurrence after curative treatment. In particular, HOMA-IR in non-diabetic patients is an extremely useful tool for predicting the recurrence risk of HCC. In order to identify the recurrence of HCC in the early stage, active surveillance is especially needed for these patients.

## Figures and Tables

**Figure 1 ijms-20-00605-f001:**
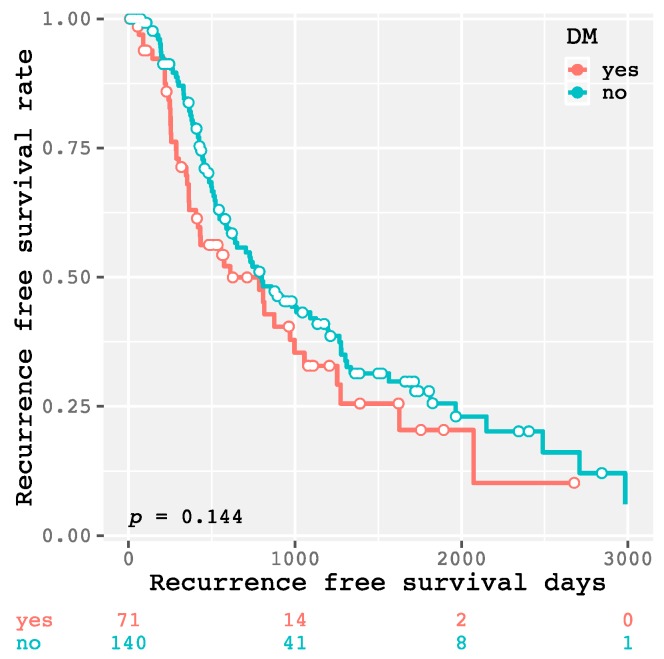
Kaplan–Meier curves for recurrence-free survival time in patients with and without diabetes mellitus.

**Figure 2 ijms-20-00605-f002:**
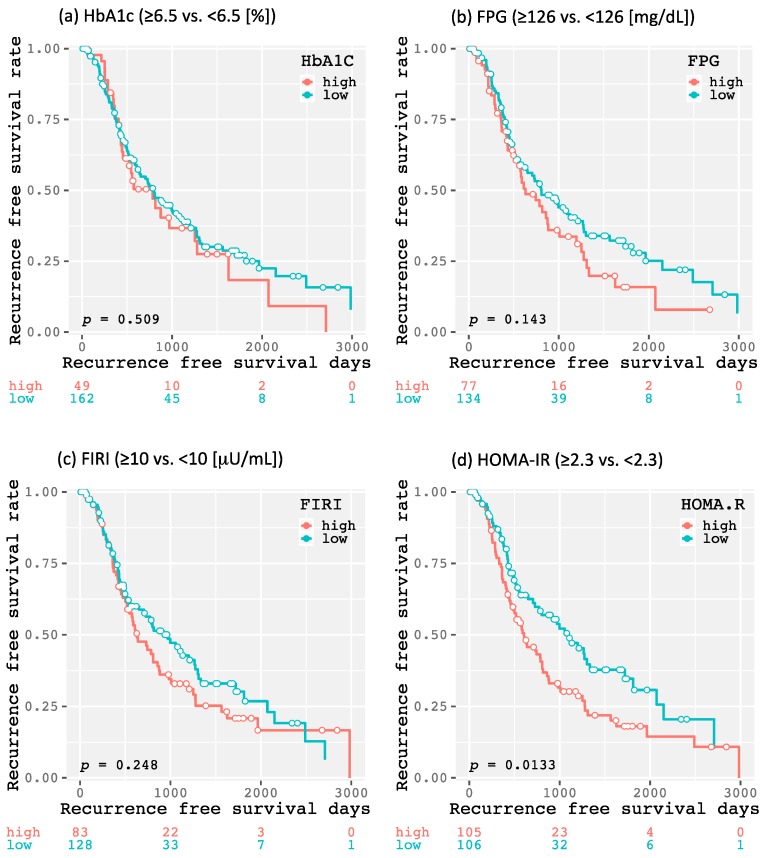
Kaplan–Meier curves for recurrence-free survival time divided into higher and lower (≥6.5 and <6.5%, respectively) hemoglobin A1c (HbA1c) groups (**a**), higher and lower (≥126 and <126 mg/dL, respectively) fasting plasma glucose (FPG) groups (**b**), higher and lower (≥10 and <10 μU/mL, respectively) fasting immunoreactive insulin (FIRI) groups (**c**), and higher and lower (≥2.3 and <2.3, respectively) homeostasis model assessment-insulin resistance (HOMA-IR) groups (**d**).

**Figure 3 ijms-20-00605-f003:**
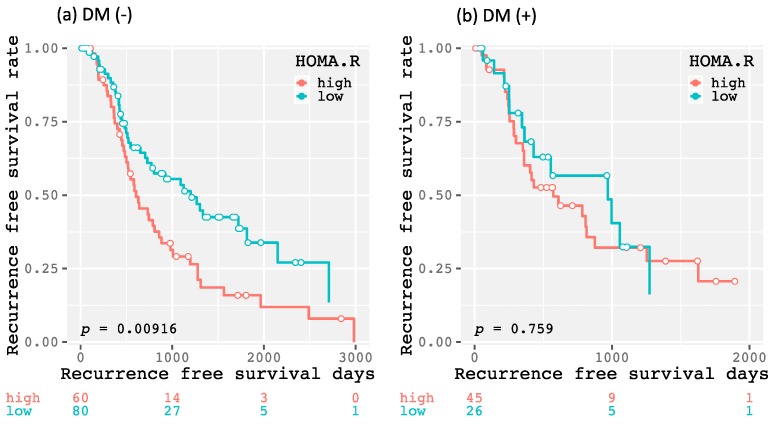
Kaplan–Meier curves for recurrence-free survival time divided into higher and lower (≥2.3 and <2.3, respectively) homeostasis model assessment-insulin resistance (HOMA-IR) groups without (**a**) or with diabetes mellitus (DM) (**b**).

**Table 1 ijms-20-00605-t001:** Baseline demographic and clinical characteristics of patients with and without diabetes mellitus.

	All Cases (*n* = 211)	DM (−) (*n* = 140)	DM (+) (*n* = 71)	*p*-Value
Sex (male/female)	148/63	96/44	52/19	0.527
Age (years)	70.6 ± 9.1	71.0 ± 8.9	69.8 ± 9.4	0.377
Etiology (HBV/HCV/others)	28/131/52	21/97/22	7/34/30	<0.001
BMI (kg/m^2^)	23.1 ± 3.1	22.5 ± 2.8	24.3 ± 3.4	<0.001
Child-Pugh score (5/6/7/8/9/10)	137/44/18/8/2/2	87/29/14/7/1/2	50/15/4/1/10	0.547
ALB (g/dL)	3.8 ± 0.5	3.7 ± 0.5	3.9 ± 0.5	0.028
ALT (IU/L)	40.4 ± 29.9	39.5 ± 24.1	42.0 ± 39.1	0.577
T-Bil (mg/dL)	1.0 ± 0.6	1.1 ± 0.6	0.9 ± 0.4	0.022
PLT (×10^4^/μL)	13.2 ± 6.3	12.8 ± 6.0	14.1 ± 17.1	0.131
PT (%)	87.1 ± 16.3	86.3 ± 15.5	88.6 ± 18.0	0.338
FPG (mg/dL)	112.2 ± 34.7	100.4 ± 15.8	135.5 ± 47.8	<0.001
FIRI (μU/mL)	12.1 ± 16.1	11. 2 ± 9.7	14.0 ± 24.2	0.243
HbA1c (%)	6.1 ± 1.2	5.6 ± 0.7	7.0 ± 1.4	<0.001
HOMA-IR	3.5 ± 6.5	2.9 ± 3.0	4.8 ± 10.3	0.047
TG (mg/dL)	102.6 ± 58.0	93.6 ± 40.4	120.2 ± 79.6	0.003
AFP (ng/dL)	552 ± 2440	626 ± 2376	409 ± 2573	0.542
PIVKA-II (mAU/mL)	6440 ± 46 580	2482 ± 17 335	14 244 ± 76 284	0.085
Stage (I/II/III/IV)	76/93/38/4	52/59/26/3	24/34/12/1	0.911
Initial treatment (resection/RFA)	99/112	63/77	36/35	0.523
Co-existing diseases (yes/no)				
Renal disease	11/200	6/134	5/66	0.513
Heart disease	27/184	14/126	13/58	0.125
Neurologic disease	12/199	6/134	6/65	0.224
Hypertension	86/125	42/98	44/27	<0.001
Hyperlipidemia	17/194	6/134	11/60	0.007
NAFLD/NASH	16/195	3/137	13/58	<0.001

Values are presented as a mean ± standard deviation. DM, diabetes mellitus; HBV, hepatitis B virus; HCV, hepatitis C virus; BMI, body mass index; ALT, alanine aminotransferase; T-Bil, total bilirubin; PLT, platelet count; PT, prothrombin time; FPG, fasting plasma glucose; FIRI, fasting immunoreactive insulin; HbA1c, hemoglobin A1c; HOMA-IR, homeostasis model assessment-insulin resistance; TG, triglyceride; AFP, alpha-fetoprotein; PIVKA-II, protein induced by vitamin K absence or antagonists-II; RFA, radiofrequency ablation; NAFLD, non-alcoholic fatty liver disease; NASH, non-alcoholic steatohepatitis.

**Table 2 ijms-20-00605-t002:** Univariate and multivariate analyses to predict hepatocellular carcinoma recurrence in all patients.

	Univariate Analysis		Multivariate Analysis	
Variable	HR (95% CI)	*p* Value	HR (95% CI)	*p* Value
Sex (male vs. female)	1.221 (0.829–1.798)	0.311		
Age (years)	0.993 (0.973–1.014)	0.525		
BMI (kg/m^2^)	1.061 (1.000–1.125)	0.049	1.047 (0.986–1.111)	0.134
Child (A vs. B/C)	1.032 (0.609–1.746)	0.908		
Albumin (g/dL)	0.939 (0.647–1.361)	0.738		
PLT (×10^4^/mL)	0.988 (0.959–1.018)	0.434		
HOMA-IR (≥2.3 vs. <2.3)	1.565 (1.095–2.239)	0.014	1.485 (1.030–2.139)	0.034
AFP (ng/dL)	1.000 (0.999–1.000)	0.503		
PIVKA-II (mAU/mL)	1.000 (1.000–1.000)	0.168		
Stage (I vs. II/III/IV)	0.913 (0.632–1.320)	0.628		
Initial treatment (RFA vs. resection)	1.103 (0.771-1.577)	0.592		

HR, hazard ratio; CI, confidence interval; BMI, body mass index; PLT, platelet count; HOMA-IR, homeostasis model assessment-insulin resistance; AFP, alpha-fetoprotein; PIVKA-II, protein induced by vitamin K absence or antagonists-II; RFA, radiofrequency ablation.

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
