# Peer review of "Homeostatic Model Assessment of Insulin Resistance for Predicting the Recurrence of Hepatocellular Carcinoma after Curative Treatment"

_ijms, 2019, doi:10.3390/ijms20030605_

Round 1

Reviewer 1 Report

I suggest to separate non-alcoholic fatty liver disease (NAFLD) from others in etiology since a lot of studies have indicated that NAFLDis are highly prevalent in type 2 diabetes mellitus (T2DM). 

Please indicate the types of DM patients (type 1 or type 2). 

The sample size of DM HCC patients was too small. 

Please specify how was the statistical analysis performed for patients  with different etiology, HBV/HCV/others. 

Please clarify when the data in table 1 was collected for each patient. Right after curative HCC treatment? 

Please clarify if any diabetes related complications, such as kidney failure, were observed in the enrolled patients.

Please clarify if diabetes of DM patients with treatments had been well managed.

Author Response

Responses to Reviewer #1:

                We wish to thank this reviewer commenting our study provides important information for the management of HCC patients. We also thank this reviewer’s constructive comments which were most helpful to improve our manuscript. We accordingly revised the manuscript as follows.

1.             I suggest to separate non-alcoholic fatty liver disease (NAFLD) from others in etiology since a lot of studies have indicated that NAFLDs are highly prevalent in type 2 diabetes mellitus (T2DM).

                Following this suggestion, we analyzed a recurrence risk in 195 patients who did not have NASH/NAFLD and showed new Supplementary Figures 1 to 3. We found that the factors associated with a disorder of glucose metabolism such as the presence of DM (P = 0.092; Supplementary Fig. 1), higher HbA1c level (P = 0.179; Supplementary Fig. 2a), FPG level (P = 0.115; Supplementary Fig. 2b), and FIRI level (P = 0.284; Supplementary Fig. 2c) were not significant predictors for HCC recurrence, but only higher HOMA-IR level predicted the recurrence (P = 0.015; Supplementary Fig. 2d). Moreover, even in non-NASH/NAFLD patients who did not have DM, higher HOMA-IR level had a significant impact on the recurrence of HCC (P = 0.011; Supplementary Fig. 3a), whereas there was no significant difference when the patients had DM (P = 0.686; Supplementary Fig 3b, lines 101-109). We believe these new findings increase the value of our manuscript. We thank your valuable suggestion.

2.             Please indicate the types of DM patients (type 1 or type 2)

                In this study, all of the diabetic patients were classified into type 2 DM. We revised the text (lines 68-69).

3.             The sample size of DM HCC patients was too small. 

                As suggested, we fully understand this is one of the limitations of the present. We therefore consider long-term, prospective studies in a larger number of patients should be performed to confirm the findings of the present study. In the revised text, we clarified these important matters (lines 176-177 and 181-183). We are grateful for your valuable advice again.

4.             Please specify how was the statistical analysis performed for patients with different etiology, HBV/HCV/others.

                Following this suggestion, we rewrote the Materials and Methods section by citing new reference (lines 226-228 and new Reference #38).Holm-Bonferroni correction was used to counteract the problem of multiple comparisons.

5.             Please clarify when the data in table 1 was collected for each patient. Right after curative HCC treatment ?

                The data shown in Table 1 were collected immediately before initial treatment. (lines 68).

6.             Please clarify if any diabetes related complications, such as kidney failure, were observed in the enrolled patients.

                According to this suggestion, we added the prevalence rate of co-existing diseases such as renal disease, heart disease, neurologic disease, hypertension, hyperlipidemia and NASH/NAFLD in revised Table 1. Among them, hypertension, hyperlipidemia and NASH/NAFLD were more often observed in patients with DM than in those without DM (lines 74-76).

7.             Please clarify if diabetes of DM patients with treatments had been well managed.

                We evaluated glucose metabolism only once immediately before initial treatment. Therefore, it is unclear whether diabetes of DM patients with treatments had been well managed. In addition, it remains unclear whether adequate DM treatment that improves both hyperglycemia and hyperinsulinemia could reduce a risk of HCC recurrence in patients with DM. We understand these are critical limitations which should be mentioned. We also consider long-term, prospective studies in a larger number of patients should be performed to confirm the findings of the present study. We emphasized these points in the revised text (lines 177-182).

Reviewer 2 Report

Previous study reported that a higher level of homeostasis model assessment-insulin resistance (HOMA-IR), an indicator of insulin resistance, is associated with an increased risk for recurrence of stage I HCC after curative treatment in HCV-positive patients. The present study demonstrated that a higher HOMA-IR level (≥2.3) can be used to predict the early recurrence of HCC, whereas the complication of DM, a higher HbA1c level (≥6.5%), higher FPG level (≥126 mg/dL), and higher FIRI level (≥10 mU/mL) were not associated with recurrence. Moreover, non-diabetic patients, but not diabetic patients, with a higher HOMA-IR level had a significantly increased risk of HCC recurrence after curative treatment. There are some concerns as listed in the following:

(1) Previous study applied multivariate analysis to reveal HOMA-IR to be independent predictors for recurrence of stage I HCC after curative RFA in HCV-positive patients (Ref.15). No multivariate analysis data were presented in the present study.

(2) Typos and others:

*L79: average (?mean) ± standard deviation

*L95-96: there was a significant difference between the higher and lower HOMA-IR groups when the patients did have? (did not have)DM (P = 0.009; Fig. 3a)

*L206: References: check all to keep one consistent format for journal name, i.e. full name or abbreviation (e.g. R1 vs. R2); page number (e.g. R4 vs. R8) 

Author Response

Responses to Reviewer #2:

                We are pleased that in the overall comments this reviewer found our study is of interest.

1.             Previous study applied multivariate analysis to reveal HOMA-IR to be independent predictors for recurrence of stage I HCC after curative RFA in HCV-positive patients (Ref.15). No multivariate analysis data were presented in the present study.

                According to this suggestion, we conducted univariate and multivariate analyses to predict HCC recurrence by Cox proportional hazards model. Univariate analysis revealed that higher body mass index (P = 0.049) and HOMA-IR level (P = 0.014) were significant risk factors of HCC recurrence. Multivariate analysis identified that higher HOMA-IR level is the only independent predictor for HCC recurrence in this study (hazard ratio, 1.485; 95% confidence interval, 1.030-2.139; P = 0.034). This finding was consistent with the result of previous report (Ref. #15) suggested by this reviewer (lines 123-129, lines 227-228, and new Table 2). We believe the results of these new analyses increase the value of our study. We thank your important suggestion.

2.             Typos and others:

*L79: average (?mean) ± standard deviation

*L95-96: there was a significant difference between the higher and lower HOMA-IR groups when the patients did have? (did not have)DM (P = 0.009; Fig. 3a)

*L206: References: check all to keep one consistent format for journal name, i.e. full name or abbreviation (e.g. R1 vs. R2); page number (e.g. R4 vs. R8) 

                According to these indications, I corrected writing errors (lines 82 and 100, and References section). We thank your detailed indications.

Round 2

Reviewer 1 Report

Authors clearly answered my questions and reorganized the data.